# Deep Learning Approach Predicts Longitudinal Retinal Nerve Fiber Layer Thickness Changes [note 1]

**DOI:** 10.3390/bioengineering12020139

**Published:** 2025-01-31

**Authors:** Jalil Jalili, Evan Walker, Christopher Bowd, Akram Belghith, Michael H. Goldbaum, Massimo A. Fazio, Christopher A. Girkin, Carlos Gustavo De Moraes, Jeffrey M. Liebmann, Robert N. Weinreb, Linda M. Zangwill, Mark Christopher

**Affiliations:** 1Hamilton Glaucoma Center and Division of Ophthalmology Informatics and Data Science, Shiley Eye Institute, Viterbi Family Department of Ophthalmology, University of California, San Diego, CA 92037, USA; jjalili@health.ucsd.edu (J.J.); ehwalker@health.ucsd.edu (E.W.); cbowd@health.ucsd.edu (C.B.); abelghith@health.ucsd.edu (A.B.); mgoldbaum@health.ucsd.edu (M.H.G.); cgirkin@health.ucsd.edu (C.A.G.); rweinreb@health.ucsd.edu (R.N.W.); lzangwill@health.ucsd.edu (L.M.Z.); 2Department of Ophthalmology and Vision Sciences, Heersink School of Medicine, University of Alabama at Birmingham, Birmingham, AL 35233, USA; massimofazio@uabmc.edu; 3Bernard and Shirlee Brown Glaucoma Research Laboratory, Department of Ophthalmology, Edward S. Harkness Eye Institute, Columbia University Medical Center, New York, NY 10032, USA; cvd2109@cumc.columbia.edu (C.G.D.M.); jml2314@cumc.columbia.edu (J.M.L.)

**Keywords:** glaucoma, deep learning, RNFL thickness prediction, optical coherence tomography, one-dimensional convolutional neural network, longitudinal OCT

## Abstract

This study aims to develop deep learning (DL) models to predict the retinal nerve fiber layer (RNFL) thickness changes in glaucoma, facilitating the early diagnosis and monitoring of disease progression. Using the longitudinal data from two glaucoma studies (Diagnostic Innovations in Glaucoma Study (DIGS) and African Descent and Glaucoma Evaluation Study (ADAGES)), we constructed models using optical coherence tomography (OCT) scans from 251 participants (437 eyes). The models were trained to predict the RNFL thickness at a future visit based on previous scans. We evaluated four models: linear regression (LR), support vector regression (SVR), gradient boosting regression (GBR), and a custom 1D convolutional neural network (CNN). The GBR model achieved the best performance in predicting pointwise RNFL thickness changes (MAE = 5.2 μm, R^2^ = 0.91), while the custom 1D CNN excelled in predicting changes to average global and sectoral RNFL thickness, providing greater resolution and outperforming the traditional models (MAEs from 2.0–4.2 μm, R^2^ from 0.94–0.98). Our custom models used a novel approach that incorporated longitudinal OCT imaging to achieve consistent performance across different demographics and disease severities, offering potential clinical decision support for glaucoma diagnosis. Patient-level data splitting enhances the evaluation robustness, while predicting detailed RNFL thickness provides a comprehensive understanding of the structural changes over time.

## 1. Introduction

Glaucoma is a progressive disease characterized by longitudinal structural changes to the eye and loss of function [1,2]. It is one of the leading causes of irreversible blindness worldwide, affecting an estimated 76 million people in 2020, with this number expected to rise to 111.8 million by 2040 due to population aging and growth [3]. The prevalence of glaucoma varies by region, with higher rates observed in populations of African and Asian descent [4,5]. Because progression is gradual, patients are often unaware of their disease until they have already suffered substantial, irreversible damage [6,7,8]. In fact, previous work suggests that more than 50% of individuals with glaucoma are unaware of their disease, which can lead to substantial vision loss before patients become symptomatic of irreversible visual impairment [7,9]. The improved detection and forecasting of structural changes is a critical need to help improve glaucoma management and preserve vision.

The current standard for glaucoma diagnosis and management includes monitoring structural changes using optical coherence tomography (OCT), primarily of the optic nerve head (ONH) region [10,11]. This imaging modality provides digital histologic reconstruction of the retina and can be used to collect reliable quantitative tissue measurements related to glaucoma including ONH parameters and retinal nerve fiber layer (RNFL) thickness [12,13]. A limitation of this approach, however, is that follow-up visits over a long period of time might be required to identify glaucoma onset and progression [14,15]. An approach that can accurately predict a patient’s future structure would help clinicians more quickly identify progressing disease and adjust the treatment appropriately. The application of artificial intelligence (AI) is one such approach.

Artificial intelligence (AI) advances are having a transformative effect on medicine [16,17]. In particular, deep learning (DL) approaches, such as convolutional neural networks (CNNs), focused on making predictions from images, have had an especially large impact on the image-intensive specialty of ophthalmology. DL models have been trained to detect eye disease (diabetic retinopathy [18], macular degeneration [19], glaucoma [20]), segmenting anatomical and disease features from images [21], and predicting visual function from imaging [2], among other applications. Across these tasks, DL models have achieved high performance, often matching or exceeding the performance of human experts [22,23,24].

Several previous studies have utilized AI and deep learning (DL) models for glaucoma diagnosis and progression detection [2,12]. Despite these advancements, a significant gap remains in the application of DL techniques for predicting longitudinal structural changes, such as RNFL thinning. Predicting these changes is crucial for the early detection of disease progression, as RNFL thinning is one of the earliest indicators of glaucomatous damage [25,26,27,28]. However, most of these studies have focused on cross-sectional data, using OCT images from a single time point to classify disease status, predict visual function, or estimate RNFL thickness [29,30]. As a result, these models are generally limited to diagnosing current disease rather than forecasting future structural changes. For instance, while some studies have attempted to predict the RNFL thickness from OCT images, these models are restricted to predicting the thickness values for the same visit rather than for future visits [29,30,31].

Only one study has attempted to predict the future RNFL thickness using longitudinal data [32]. While this work represents an important step forward, certain limitations remain. The study included different visits from the same patients in both training and testing sets, which may have led to data leakage, potentially inflating the model performance and limiting the generalizability to unseen patient data. Furthermore, the prediction was limited to five regional RNFL values, restricting the analysis’s granularity. Predicting the entire 768-point RNFL thickness vector is crucial for capturing localized retinal changes, offering greater precision for monitoring disease progression and informing clinical decision making. Therefore, there remains a need for models capable of predicting the full RNFL thickness vector for future visits, with robust evaluation using patient-level data splitting to ensure reliable generalization.

The success of DL approaches across various problems suggests their potential suitability for longitudinal RNFL predictions. However, to date, there has been relatively little research focused on the DL methods for this task. The objective of this study is to leverage a large longitudinal dataset to develop AI models capable of predicting the structural changes in a cohort of healthy and glaucoma eyes. Specifically, we aim to predict future circumpapillary RNFL thickness based on previous ONH OCT imaging. Our approach predicts the entire 768-point RNFL thickness vector, providing a more detailed and granular assessment of retinal changes compared to prior studies that were limited to only five regions.

The rest of the paper is organized as follows. Section 2 describes our datasets, preprocessing, and predictive models. Section 3 presents the results of evaluating our predictive models, Section 4 discusses these results and places them in context with other recent work, and Section 5 concludes the work.

## 2. Materials and Methods

### 2.1. Datasets

The datasets used for this analysis were collected as part of two longitudinal glaucoma studies: the Diagnostic Innovations in Glaucoma Study (DIGS, clinicaltrials.gov identifier: NCT00221897) and the African Descent and Glaucoma Evaluation Study (ADAGES, clinicaltrials.gov identifier: NCT00221923) [33]. These studies have been detailed previously [34]. The inclusion criteria required the participants to have a best-corrected Snellen visual acuity of 20/40 or better and at least two consecutive reliable standard automated perimetry visual field (VF) tests at the study entry. Glaucoma was defined as eyes with repeated abnormal VF results. Healthy eyes were identified as those with no abnormal VF results and normal-appearing optic discs based on a fundus image review. Suspect eyes were defined as those with glaucomatous optic neuropathy observed on fundus imaging but without detectable VF damage.

The participants underwent semi-annual OCT imaging using a Spectralis device (Heidelberg Engineering GmbH, Heidelberg, Germany). For this work, OCT imaging was captured using the optic nerve head radial—circle (ONHRC) scan pattern that captured 24 radial scans and 3 circle scans at diameters of 3.5 mm, 4.1 mm, and 4.5 mm, centered on the ONH. The retinal nerve fiber layer (RNFL) segmentation was performed using built-in software provided by the manufacturer, which generated 768 equally spaced RNFL thickness measurements around the circle scans, along with global and sectoral RNFL thickness averages. Sectoral thickness measurements included temporal, temporal—superior, temporal—inferior, nasal, nasal—superior, and nasal—inferior. Figure 1 provides an example OCT circle scan along with the segmented RNFL thickness and corresponding sectors. OCT images and segmentations were evaluated for quality by using the UC San Diego Imaging Data Evaluation and Analysis (IDEA) Reading Center according to standard protocols [35]. OCT images with poor signal quality, imaging artifacts, and/or segmentation errors that could not be manually corrected were excluded from further analysis.

VF testing was performed using the 24-2 Swedish interactive thresholding algorithm (SITA standard, Humphrey Field Analyzer II; Carl Zeiss Meditec, Inc., Dublin, CA, USA) [33]. VFs with more than 33% fixation losses, false-negative errors, or false-positive errors were excluded following the UC San Diego Visual Field Assessment Center (VisFACT) established protocols. Key metrics such as mean deviation (MD), pattern standard deviation (PSD), visual field index (VFI), and individual test point deviations were extracted.

### 2.2. Input–Target Preprocessing

The objective of this study was to predict the future RNFL thicknesses based on prior OCT imaging. For this analysis, OCT imaging triplets were constructed. Each triplet consisted of three OCT ONHRC scans obtained during different visits, with the time between the visits ranging from 6 to 18 months. The first two visits served as inputs, while the third visit provided the RNFL thickness values used as the target for prediction. Our study employed a strict patient-level data split, ensuring no overlap among the training, validation, and test sets. Our OCT imaging dataset included semi-annual visits for most patients, with some instances where multiple ONHRC scans were captured during a single visit. As a result, multiple valid permutations of OCT imaging triplets could be constructed for a single eye over a given time period. For model training, all the possible triplet permutations were included to maximize the training dataset size. In contrast, for model testing, a single OCT ONHRC scan was randomly selected from each visit to construct unique triplets. This approach ensured that no conflicting triplets (i.e., identical input scans with different target scans) would impact the results. Figure 1 illustrates a representative Spectralis circle scan of a glaucoma patient, segmented with ILM, RNFL, and BM layers, and a visualization of the corresponding 768-element RNFL thickness vector. Figure 2 and Appendix A provide detailed schematics of the methodology used to construct the input–target data series and apply the deep learning model.

The two specific outputs of our models were a 768-element vector estimating the RNFL thickness in a circle around the ONH and a 7-element vector estimating the global and sectoral RNFL thickness averages. Small errors in the segmentation or missing values were addressed by filling values with neighboring measurements and truncating the RNFL thickness to 300 μm [36,37]. Following preprocessing, our training dataset (train and validation) encompassed 691 input–target series from 205 participants (357 eyes), while the testing set included 126 instances from 46 participants (80 eyes).

### 2.3. Predictive Methods

We evaluated several different model types for predicting the RNFL thickness including linear regression (LR), support vector regression (SVR) [38], gradient boosting regression (GBR) [39], and a custom 1D convolutional neural network (CNN) [40]. For all the models, the input consisted of the RNFL thickness vectors derived from OCT imaging. Specifically, two 768-element RNFL thickness vectors from the first and second visits, representing the baseline and follow-up measurements, were used as inputs to predict either a 768-element RNFL thickness vector or a 7-element vector summarizing the global and sectoral averages.

For the linear regression, SVR, and GBR models, the two 768-element RNFL thickness vectors were concatenated into a single 1536-element feature vector, representing the complete RNFL profile from both visits. For the 1D-CNN, the same two RNFL thickness vectors were stacked to create a two-channel input of size (2 × 768). This format preserved the spatial arrangement of the RNFL thickness measurements along the circular scan, allowing the CNN to leverage the structural information inherent in the data.

Our custom 1D CNN consisted of four convolutional layers interspersed with ReLU and max pooling layers. These were followed by fully connected layers and a final output layer that predicted either 768 RNFL thickness values or a 7-element vector predicting the global and sectoral RNFL thickness averages. Appendix A provides a schematic of our custom 1D CNN. The CNN was trained using an Adam optimizer with a learning rate of 0.001 and a mean squared error loss function for 100 epochs.

For all the cases, we partitioned the data by participant, using 80% for training/validation and 20% for testing. This guaranteed no overlap between the sets—no data from a training participant were present in the testing set and vice versa. For the CNN training, data augmentation was applied by randomly shifting thickness vectors (±5 pixels) during training.

### 2.4. Model Evaluation

We evaluated the models for their accuracy in predicting both the full 768-element RNFL thickness vector and the 7 global and sectoral averages (G, T, TS, TI, N, NS, NI) in the next visit. The model’s performance was quantitatively evaluated using the mean absolute error (MAE), mean relative error (MRE), and R2. We also computed these metrics comparing the RNFL thicknesses from the second input OCT to the target RNFL thicknesses. The models were also evaluated as a function of the participant demographics (age, race) and disease status and severity as measured by using 24-2 VF MD.

## 3. Results

This study included 1744 images from 251 subjects and 437 eyes (Table 1). Of the initial 502 eyes, 65 were excluded due to the poor image quality, segmentation errors, or insufficient consecutive OCT visits. These images were assembled into 691 input–target series used for training/validation and 126 input–target series for testing. The average age of the participants at the bassline visit in this study was 68.8 years, and female participants (n = 136, 54.2%) outnumbered male participants (n = 115, 45.8%). The majority of the participants self-identified as either Black/African American (n = 82, 32.7%) or White (n = 139, 55.4%), with a smaller proportion identifying as Asian (n = 25, 10.0%) or other/unknown (n = 5, 2.0%). The dataset included 196 healthy eyes (24.0%), 263 suspect eyes (32.2%), and 358 glaucoma eyes (43.8%). The average global RNFL thickness was 86.4 μm for healthy eyes, 72.8 μm for suspect eyes, and 74.5 μm for glaucoma eyes. No significant differences in patient demographics, glaucoma status, or RNFL thickness were observed between the training/validation and test sets. Table 1 summarizes the demographic and clinical characteristics of the dataset, including the patient age, gender, race, glaucoma status, and RNFL thickness at the baseline visit.

In predicting the full 768-element RNFL thickness vector, the GBR model achieved the best performance (MAE [95% CI] = 5.2 μm [4.8–5.6], MRE = 9.4% [6.7–12.1], R2 = 0.91), followed by the 1D-CNN (MAE = 6.7 μm [6.3–7.1], MRE = 11.5% [9.7–13.4], R2 = 0.89). Both GBR and 1D-CNN performed significantly better (*p* < 0.05) than either the SVR or LR models (Table 2 and Figure 3).

In predicting the global and sectoral RNFL thickness averages, the 1D-CNN model surpassed the others in predicting the thickness averages, achieving the following MAEs: global = 2.4 μm [1.9–2.8], temporal = 2.0 μm [1.7–2.4], temporal superior = 3.6 μm [2.9–4.4], temporal inferior = 4.2 μm [3.0–5.0], nasal = 3.0 μm [2.4–3.5], nasal superior = 3.6 μm [3.1–4.2], and nasal inferior = 3.8 μm [3.1–4.5] (Table 3). The 1D-CNN also achieved R2 ranging from 0.94 to 0.98 in predicting these thickness averages. Figure 4 provides scatterplots comparing the 1D-CNN global and sectoral predictions to the ground truth.

The performance of the 1D-CNN was also evaluated as a function of patient demographics and disease status. Across the sexes, the model achieved comparable or slightly lower MAEs in the male compared to female participants (global MAEs of 2.2 μm vs. 2.5 μm, *p* = 0.56), but the differences were not statistically significant for any global or sectoral predictions (*p*-values ≥ 0.304). To evaluate the model across self-reported races, we considered only two groups of participants (Black/African American and White) because they represented the majority of our cohort. The model had consistent performance across Black/African American and White participants (global MAEs of 2.1 μm vs. 2.5 μm, *p* = 0.40). The largest difference was for the nasal superior sector (MAE 4.4 μm vs. 3.3 μm, *p* = 0.08); however, no difference reached statistical significance.

The performance of the 1D-CNN model was analyzed across different disease statuses and glaucoma severity levels. Comparing the glaucoma to non-glaucomatous participants, the model showed similar global MAEs (2.4 μm vs. 2.3 μm, *p* = 0.848) and for the suspect participants (2.4 μm vs. 2.4 μm, *p* = 0.985) with no statistically significant differences. However, some slightly larger differences emerged in specific sectors. In the temporal inferior (TI) sector, the glaucoma participants had a higher MAE (5.4 μm vs. 3.2 μm, *p* = 0.108) compared to the non-glaucomatous participants. Similarly, in the nasal inferior (NI) sector, the MAE for the glaucoma participants was higher (4.3 μm vs. 3.1 μm, *p* = 0.166) compared to the suspect participants, though none of these differences reached statistical significance.

When evaluating the glaucoma severity, the participants with moderate to advanced glaucoma showed higher MAEs in certain sectors, though the differences were not statistically significant (*p*-values ≥ 0.125). For example, in the temporal inferior (TI) sector, the moderate to advanced glaucoma participants had a higher MAE (6.8 μm vs. 4.4 μm, *p* = 0.125). In the global sector, the MAE for mild glaucoma was lower (2.2 μm vs. 3.0 μm, *p* = 0.215), but this difference did not reach statistical significance (Table 3 and Table 4).

## 4. Discussion

In this study, we developed and evaluated deep learning models, specifically a custom 1D-CNN, for predicting the longitudinal RNFL thickness changes based on OCT imaging. To our knowledge, this is one of the first studies to leverage DL models to predict the future RNFL thickness, using previous RNFL measurements to predict the structural changes over time. These results highlight the potential for AI to play a pivotal role in early glaucoma detection and disease management. Our results demonstrate that the custom 1D-CNN outperforms the traditional machine learning approaches employing LR and SVR approaches in predicting both full 768-element RNFL thickness vectors and sectoral averages. Specifically, the 1D-CNN achieved the best overall performance in predicting the global and sectoral RNFL thicknesses, with MAEs ranging from 2.03 μm to 4.16 μm. Additionally, the model demonstrated strong R^2^ values, ranging from 0.94 to 0.98 across all the sectors, underscoring its robustness and high predictive accuracy.

In evaluating the performance of the 1D-CNN across various demographic and ocular characteristics, such as sex, race, disease status, and glaucoma severity, the model consistently demonstrated strong predictive capabilities. However, slight deviations were noted in certain sectors, particularly in the temporal inferior (TI) and nasal inferior (NI) regions, which were more pronounced in the participants with moderate to advanced glaucoma. These regions are known for having more variability in RNFL thickness measurements, especially in the advanced disease stages where the thinning is more pronounced. This finding suggests that while the 1D-CNN can handle most RNFL thickness predictions effectively, improvements in handling these specific sectors, especially in the more advanced disease, could enhance the model’s robustness. For future iterations, incorporating a more tailored architecture that specifically addresses these challenging sectors could improve the accuracy. Additionally, the results across race and sex groups indicate that the model generalizes well across different demographic groups, though further validation on more diverse populations is warranted to ensure broader applicability.

The comparison of the RNFL thickness predictions from the GBR and 1D-CNN models against the true RNFL thickness vector across the normal, suspect, and glaucoma participants in Figure 3 shows that both models closely follow the true RNFL values, demonstrating their ability to capture the variations in the future RNFL thickness vectors. Although the 1D-CNN shows slightly better alignment in certain regions, particularly in the suspect and glaucoma cases, it struggles to accurately capture rapid changes (high-frequency components) within the thickness variations. This suggests that a denser 1D-CNN architecture may be needed to improve its ability to predict these rapid fluctuations more effectively, especially when forecasting the full 768-element RNFL thickness vector.

The scatterplots for the 1D-CNN model predictions of global and sectoral RNFL thickness (Figure 4) demonstrate a strong correlation between the predicted and true RNFL values across all the regions, indicating that the model performs well in capturing the RNFL thickness in most sectors. However, slight deviations from the true thickness values are observed in the superior and inferior regions, particularly in the temporal inferior and nasal inferior sectors, where the model shows less precise predictions. These regions tend to display a wider range of RNFL thickness values, making them more challenging to predict accurately compared to the nasal, temporal, and global sectors.

The ability to predict the longitudinal RNFL thickness changes is particularly impactful for early glaucoma diagnosis and monitoring the disease progression. Glaucoma is often asymptomatic in its early stages, and patients are typically unaware of vision loss until significant and irreversible damage has occurred. By using DL models to forecast the RNFL thinning, clinicians could potentially identify patients at higher risk of progression before significant functional damage occurs. OCT imaging is already a key component of glaucoma management and integrating predictive models like those described here into standard clinical workflows could provide ophthalmologists with valuable insights into a patient’s future risk of disease progression.

While several previous studies have used AI and DL models for glaucoma diagnosis and progression detection, the majority have focused on cross-sectional data, i.e., using OCT images from a single time point to classify disease status and predict visual function or RNFL thickness [2,12,29,30]. Thus, these models have generally been limited to diagnosing the current disease rather than forecasting the future structural changes. In contrast, our study focuses on predicting the longitudinal changes in the RNFL thickness, offering a more forward-looking approach to disease management. Only one other study has attempted to forecast the future RNFL thickness using longitudinal data. Sedai et al. [32] developed a model to predict the global and sectoral cpRNFL thickness based on three prior visits, incorporating 3D-OCT data, previous RNFL values, clinical test results, and demographic information, achieving MAEs between 1.8 ± 1.8 μm and 3.1 ± 2.5 μm for healthy and glaucoma patients, respectively. In contrast to our approach, their approach involved training on the first “N” visits and testing on later visits from the same patients, raising concerns about the potential data leakage, as data from a given patient could appear in both the training and test data. Our study used a strict patient-level data split, ensuring no overlap between the training and test sets. Additionally, while Sedai et al. predicted only five regional values, our 1D-CNN model forecasts the entire 768-element RNFL thickness vector, offering a more granular resolution to predict the regional changes, and potentially reveal the clinically relevant patterns that global or sectoral averages might overlook. Incorporating additional features, such as OCT volume data, could further enhance our model’s predictive capabilities, particularly in challenging regions like the temporal inferior and nasal inferior sectors.

Despite the strong performance of the 1D-CNN model, there are several limitations. First, while our model performed well overall, larger prediction errors were observed in specific sectors such as the temporal inferior (TI) and nasal inferior (NI) regions. These regions are known to be more variable and prone to measurement errors in OCT imaging, particularly in advanced glaucoma cases where the RNFL thinning is more pronounced and a RNFL floor, where further thinning is not detectable, is reached. Future work could focus on enhancing the model’s performance in these challenging sectors by incorporating additional imaging data or refining the CNN architecture. Another limitation is that our study was conducted using data drawn from two longitudinal studies, DIGS and ADAGES. While these datasets provide high-quality, standardized imaging and clinical data, the generalizability of the model to other populations or imaging devices remains unclear. The data on DIGS and ADAGES patients were collected across three regional distinct sites, but the cohort was largely limited to Black/African American and White patients in the United States. Future studies should aim to validate these findings in diverse populations and across different OCT platforms to ensure broader clinical applicability. Encouragingly, our approach performed similarly across the tested racial groups. Lastly, while this study focuses on structural predictions, future work could explore integrating functional measures such as VF data. Predicting both structural and functional outcomes simultaneously would provide a more comprehensive tool for managing glaucoma progression and could lead to more personalized and effective treatment strategies.

## 5. Conclusions

In summary, this study demonstrates the utility of deep learning models, particularly our custom 1D-CNN approach, in predicting the longitudinal RNFL thickness changes in glaucoma patients. Our models were able to accurately predict the RNFL changes across patient demographics (race and age) as well as disease severity. They were also able to provide high-resolution (i.e., 768-point) predictions surrounding the ONH. The ability to forecast the future RNFL thinning has significant clinical implications for earlier detection and intervention in glaucoma management. While additional work is needed to optimize the model’s performance in specific retinal sectors and validate its applicability in broader clinical settings, our results represent a promising step toward more proactive and personalized glaucoma care.

## Figures and Tables

**Figure 1 bioengineering-12-00139-f001:**
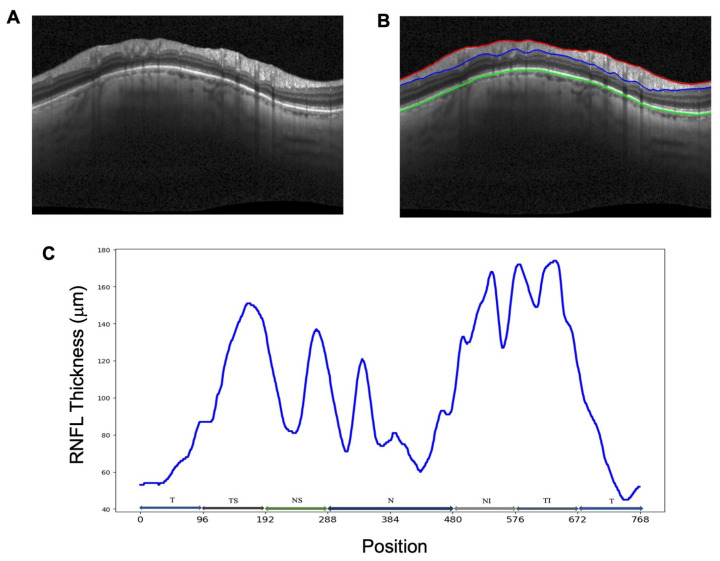
(**A**) Original Spectralis circle scan of a glaucoma patient. (**B**) Circle scan with inner limiting membrane (ILM, red), retinal nerve fiber layer (RNFL, blue), and Bruch’s membrane (BM, green) segmentations shown. (**C**) Visualization of 768-element RNFL thickness vector.

**Figure 2 bioengineering-12-00139-f002:**
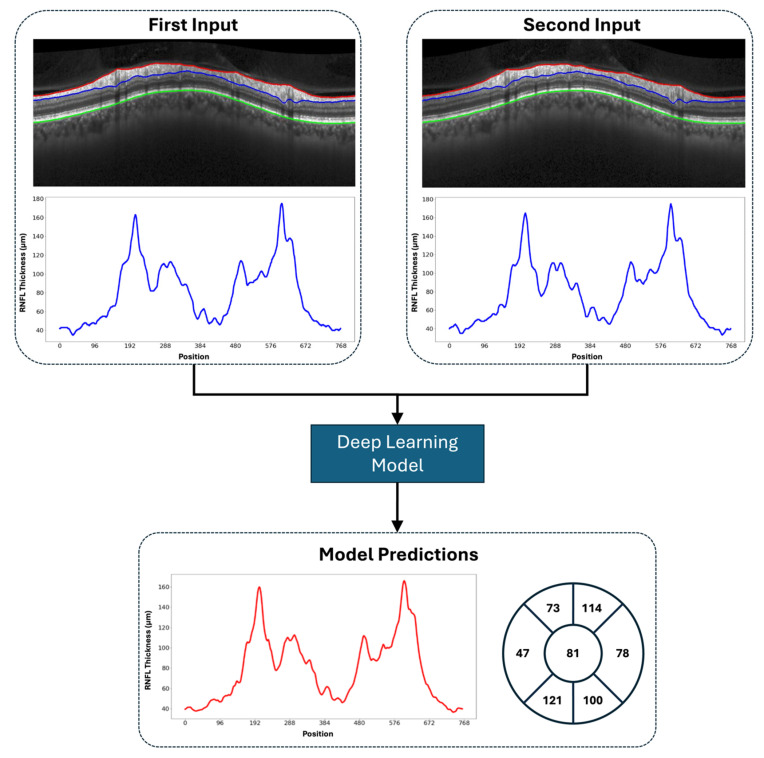
Schematic representation of our method for predicting either the future 768-element RNFL thickness vector or the 7-element vector summarizing global and sectoral averages, based on two previous 768-element RNFL thickness vectors segmented from OCT imaging.

**Figure 3 bioengineering-12-00139-f003:**
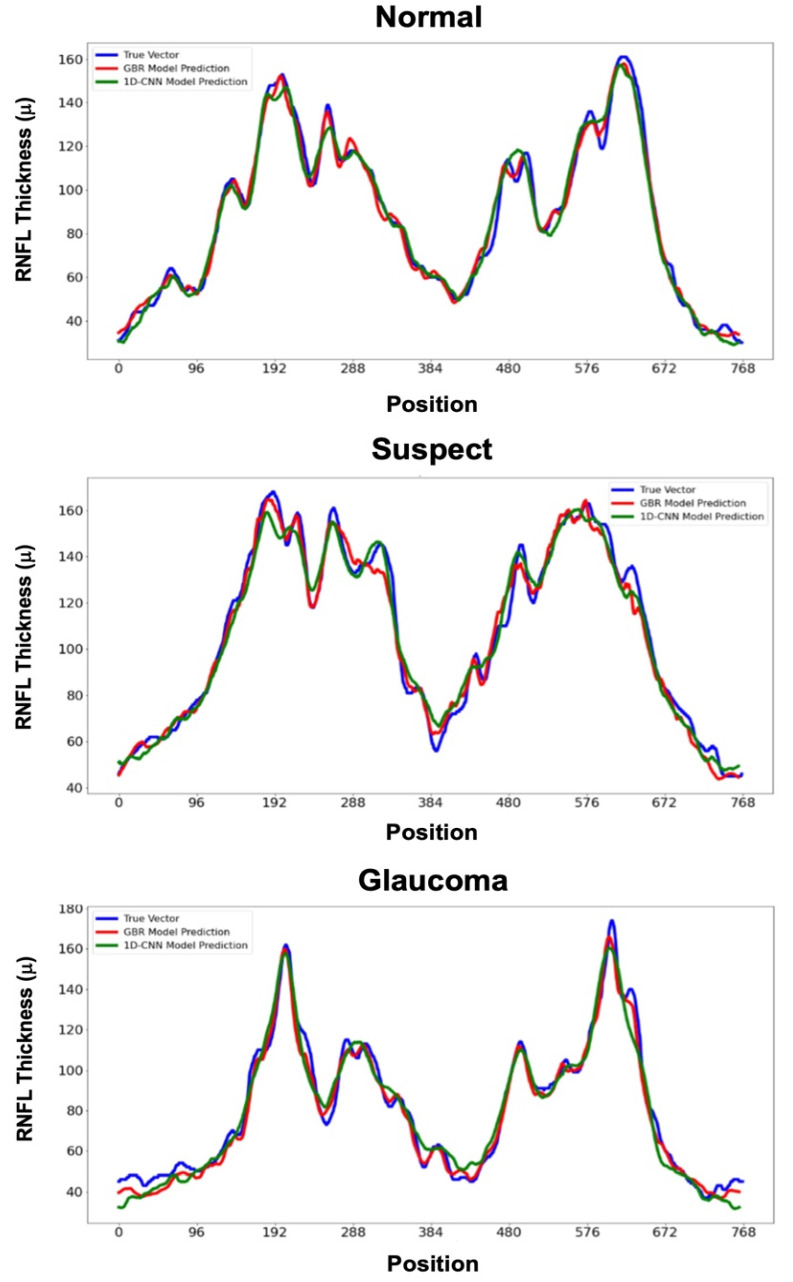
True and predicted RNFL thickness values across 768 ONH circle positions for healthy (**top**), suspect (**center**), and glaucoma (**bottom**) eyes, illustrating that both models closely follow the true RNFL values and effectively capture variations in future RNFL thickness.

**Figure 4 bioengineering-12-00139-f004:**
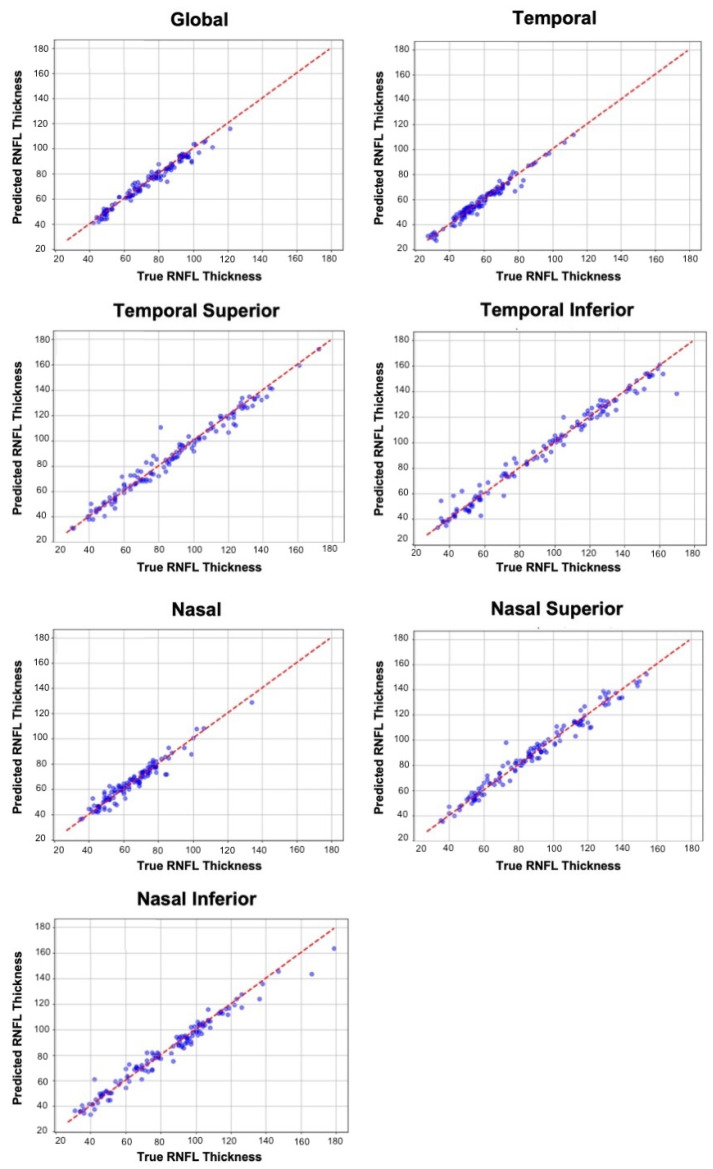
Scatterplots comparing true and predicted RNFL thickness for global and sectoral averages, showing a strong correlation across all regions.

**Table 1 bioengineering-12-00139-t001:** Summary of the dataset used for training, validation, and testing the predictive models. *p*-values compare the train and validation vs. test sets. Ranges indicate 95% confidence intervals.

	Overall	Train and Validation	Test	*p*-Value
Patients, n	251	205	46	
Eye, n	437	357	80	
Images, n	1744	1444	300	
Input–Target Series	817	691	126	
Age at the First Visit, years	68.8 (67.4, 70.2)	69.2 (67.6, 70.8)	67.2 (64.0, 70.5)	0.30
Gender, n (%)				
Female	136 (54.2%)	113 (55.1%)	23 (50.0%)	0.62
Male	115 (45.8%)	92 (44.9%)	23 (50.0%)	
Race, n (%)				
Asian	25 (10.0%)	23 (11.2%)	2 (4.3%)	0.65
Black/African American	82 (32.7%)	65 (31.7%)	17 (37.0%)	
White	139 (55.4%)	112 (54.6%)	27 (58.7%)	
Other/Unknown or Not Reported	5 (2.0%)	5 (2.5%)	0 (0.0%)	
Diagnosis at the First Visit, n (%)				
Glaucoma	358 (43.8%)	306 (44.3%)	52 (41.3%)	>0.99
Normal	196 (24.0%)	160 (23.2%)	36 (28.6%)	
Suspect	263 (32.2%)	225 (32.6%)	38 (30.2%)	
Glaucoma Severity, n (%)				
24-2 VF MD > −6.0 dB	235 (65.6%)	208 (68.0%)	27 (51.9%)	0.613
24-2 VF MD ≤ −6.0 dB	123 (34.4%)	98 (32.0%)	25 (48.1%)	
RNFL Thickness at the First Visit, μm				
Global	76.8 (74.7, 79.0)	77.2 (74.8, 79.6)	75.2 (70.1, 80.3)	0.48
Temporal	59.6 (57.6, 61.5)	59.8 (57.7, 62.0)	58.3 (53.8, 62.8)	0.55
Temporal Superior	93.6 (90.0, 97.2)	94.1 (90.1, 98.1)	91.3 (82.8, 99.8)	0.57
Temporal Inferior	100.4 (95.6, 105.2)	101.5 (96.2, 106.8)	95.5 (84.2, 106.8)	0.35
Nasal	67.5 (65.5, 69.5)	67.7 (65.5, 69.9)	66.6 (61.9, 71.2)	0.68
Nasal Superior	91.6 (88.3, 94.9)	91.9 (88.2, 95.5)	90.7 (82.9, 98.5)	0.79
Nasal Inferior	86.7 (83.4, 90.0)	87.0 (83.4, 90.7)	85.2 (77.4, 93.0)	0.68

**Table 2 bioengineering-12-00139-t002:** Summary of the full 768-element RNFL thickness vector predictions for all models.

Model	MAE (μm)	MRE (%)	R2
LR	14.3 (13.5, 15.2)	24.3 (20.7, 27.9)	0.31
SVR	11.3 (10.3, 12.3)	19.2 (14.9, 23.3)	0.70
GBR	5.2 (4.8, 5.6)	9.4 (6.7, 12.1)	0.91
1D-CNN	6.7 (6.3, 7.1)	11.5 (9.7, 13.4)	0.89

**Table 3 bioengineering-12-00139-t003:** Summary of global and sectoral RNFL thickness predictions for all models.

Model	Global			Temporal			Temporal Superior			Temporal Inferior		
	MAE (μm)	MRE (%)	R2	MAE (μm)	MRE (%)	R2	MAE (μm)	MRE (%)	R2	MAE (μm)	MRE (%)	R2
**LR**	5.0 (4.1, 5.8)	7.5 (5.8, 9.2)	0.85	5.1 (4.3, 5.9)	9.6 (7.7, 11.4)	0.85	9.3 (7.6, 11.0)	13.4 (9.5, 17.3)	0.84	10.7 (9.1, 12.4)	14.1 (11., 17.2)	0.88
**SVR**	2.8 (2.2, 3.4)	3.9 (3.2, 4.6)	0.96	3.4 (2.3, 4.4)	6.2 (4.4, 8.0)	0.96	7.4 (5.4, 9.3)	9.7 (6.8, 12.5)	0.92	6.5 (5.0, 8.1)	9.5 (6.6, 12.4)	0.96
**GBR**	4.1 (3.5, 4.7)	5.9 (5.0, 6.8)	0.93	2.7 (2.1, 3.2)	4.6 (3.9, 5.3)	0.95	4.9 (4.1, 5.6)	6.5 (5.2, 7.8)	**0.97**	4.9 (4.0, 5.7)	6.2 (4.5, 7.8)	0.97
**1D-CNN**	**2.4** **(1.9, 2.8)**	**3.3** **(2.7, 3.8)**	**0.97**	**2.0** **(1.7, 2.4)**	**3.8** **(3.1, 4.5)**	**0.97**	**3.6** **(2.9, 4.4)**	**4.6** **(3.4, 5.9)**	**0.97**	**4.2** **(3.0, 5.0)**	**5.4** **(3.6, 7.3)**	**0.98**
**Model**	**Nasal**			**Nasal Superior**			**Nasal Inferior**					
	**MAE (μm)**	**MRE (%)**	**R2**	**MAE (μm)**	**MRE (%)**	**R2**	**MAE (μm)**	**MRE (%)**	**R2**			
**LR**	6.7 (5.8, 7.7)	11.3 (9.2, 13.5)	0.72	9.5 (8.2, 10.8)	13.1 (10.6, 15.7)	0.82	9.2 (7.4, 11.0)	13.5 (8.7, 18.4)	0.84			
**SVR**	3.8 (2.6, 5.0)	5.9 (4.4, 7.3)	0.91	6.7 (5.0, 8.3)	8.4 (6.4, 10.4)	0.92	7.4 (5.4, 9.5)	10.2 (7.8, 12.6)	0.90			
**GBR**	3.7 (2.90, 4.47)	6.0 (4.8, 7.1)	0.90	4.5 (3.8, 5.2)	5.5 (4.6, 6.5)	0.96	4.2 (3.4, 5.0)	5.8 (4.4, 7.3)	0.97			
**1D-CNN**	**3.0** **(2.4, 3.5)**	**4.9** **(3.9, 5.8)**	**0.94**	**3.6** **(3.1, 4.2)**	**4.4** **(3.5, 5.1)**	**0.97**	**3.8** **(3.1, 4.5)**	**5.4** **(3.8, 7.0)**	**0.97**			

**Table 4 bioengineering-12-00139-t004:** Summary of global and sectoral RNFL thickness predictions of 1D-CNN model across different gender, race, disease status, and glaucoma severity.

Gender									
Sector	Male					Female			
	MAE		MRE	R2		MAE	MRE		R2
G	2.2 (1.7, 2.8)		3.0 (2.3, 3.8)	0.98		2.5 (1.9, 3.2)	3.6 (2.8, 4.4)		0.97
T	1.9 (1.4, 2.4)		3.5 (2.6, 4.4)	0.97		2.2 (1.6, 2.7)	4.2 (3.1, 5.2)		0.96
TS	3.6 (2.6, 4.6)		4.4 (2.7, 6.1)	0.97		3.6 (2.5, 4.7)	4.9 (3.1, 6.7)		0.98
TI	3.6 (2.1, 5.2)		5.1 (2.6, 7.5)	0.99		4.8 (3.1, 6.6)	5.9 (3.3, 8.5)		0.95
N	3.0 (2.3, 3.7)		4.9 (3.6, 6.2)	0.94		2.9 (2.2, 3.7)	4.9 (3.5, 6.3)		0.93
NS	3.6 (2.9, 4.3)		4.1 (3.0, 5.2)	0.97		3.6 (2.7, 4.5)	4.7 (3.5, 5.9)		0.99
NI	3.4 (2.5, 4.4)		5.0 (2.8, 7.1)	0.98		4.2 (3.1, 5.3)	5.8 (3.6, 8.1)		0.97
**Race**									
**Sector**	**White**					**Black/African American**			
	**MAE**		**MRE**	**R2**		**MAE**	**MRE**		**R2**
G	2.5 (2.0, 3.1)		3.5 (2.8, 4.2)	0.97		2.1 (1.4, 2.9)	3.0 (2.1, 4.0)		0.99
T	2.1 (1.6, 2.6)		3.6 (2.8, 4.4)	0.95		2.1 (1.4, 2.8)	4.5 (3.4, 5.7)		0.97
TS	3.8 (2.9, 4.8)		4.7 (3.1, 6.3)	0.98		3.5 (2.2, 4.8)	4.9 (2.8, 7.0)		0.99
TI	4.8 (3.3, 5.9)		6.7 (4.4, 9.0)	0.98		3.4 (1.6, 5.3)	3.7 (0.7, 6.8)		0.99
N	3.0 (2.3, 3.6)		4.8 (3.7, 6.0)	0.94		2.6 (1.8, 3.5)	4.3 (2.8, 5.9)		0.95
NS	3.3 (2.6, 4.0)		4.1 (3.0, 5.1)	0.98		4.4 (3.4, 5.4)	5.2 (3.8, 6.7)		0.95
NI	4.0 (3.0, 5.0)		6.2 (4.2, 8.3)	0.97		3.5 (2.3, 4.8)	4.2 (1.6, 6.8)		0.98
**Disease Status**									
**Sector**	**Normal**			**Suspect**			**Glaucoma**		
	**MAE**	**MRE**	**R2**	**MAE**	**MRE**	**R2**	**MAE**	**MRE**	**R2**
G	2.3 (1.2, 3.3)	2.6 (1.2, 4.0)	0.95	2.4 (1.4, 3.4)	3.8 (2.5, 5.1)	0.96	2.4 (1.7, 3.1)	3.3 (2.4, 4.2)	0.95
T	2.0 (1.1, 2.9)	3.0 (1.4, 4.5)	0.97	2.4 (1.5, 3.3)	5.1 (3.6, 6.5)	0.96	1.8 (1.2, 2.4)	3.3 (2.3, 4.3)	0.95
TS	3.55 (1.85, 5.25)	3.59 (0.69, 6.49)	0.97	4.09 (2.45, 5.74)	5.52 (2.82, 8.20)	0.97	3.3 (2.1, 4.4)	4.5 (2.7, 6.4)	0.99
TI	3.2 (0.5, 6.0)	2.9 (−1.7, 7.4)	0.97	3.7 (1.3, 6.1)	6.6 (2.5, 10.8)	0.97	5.4 (3.6, 7.2)	6.1 (3.2, 9.0)	0.99
N	3.4 (2.1, 4.7)	5.0 (2.6, 7.4)	0.93	2.9 (1.6, 4.1)	4.8 (2.5, 7.0)	0.91	2.7 (1.9, 3.6)	4.9 (3.3, 6.4)	0.88
NS	4.0 (2.6, 5.3)	3.5 (1.6, 5.4)	0.98	3.6 (2.3, 4.9)	4.9 (3.0, 6.8)	0.94	3.4 (2.5, 4.2)	4.5 (3.2, 5.7)	0.98
NI	4.0 (2.1, 5.8)	3.9 (0.6, 7.2)	0.96	3.1 (1.4, 4.8)	5.1 (2.1, 8.1)	0.98	4.3 (3.1, 5.6)	6.6 (4.4, 8.8)	0.94
**Disease Severity**									
**Sector**	**Mild Glaucoma**					**Moderate to Advanced Glaucoma**			
	**MAE**		**MRE**	**R2**		**MAE**	**MRE**		**R2**
G	2.2 (1.0, 3.3)		3.0 (1.7, 4.3)	0.88		3.0 (1.9, 4.1)	3.7 (2.5, 4.9)		0.96
T	1.7 (1.1, 2.3)		3.1 (2.0, 4.2)	0.96		1.9 (1.4, 2.4)	3.5 (2.5, 4.5)		0.94
TS	3.9 (2.7, 5.1)		5.4 (2.9, 7.8)	0.98		3.0 (1.9, 4.3)	4.2 (1.9, 6.6)		0.99
TI	4.4 (1.1, 7.7)		6.9 (2.7, 11.0)	0.99		6.8 (3.7, 10.0)	6.7 (2.8, 10.7)		0.99
N	2.7 (1.3, 4.1)		4.4 (1.8, 7.0)	0.88		3.1 (1.9, 4.3)	5.6 (3.2, 7.9)		0.95
NS	3.4 (2.0, 4.7)		5.1 (3.2, 7.0)	0.99		3.5 (2.3, 4.7)	4.1 (2.4, 5.9)		0.96
NI	4.8 (2.1, 7.4)		6.8 (2.4, 11.1)	0.97		5.3 (2.9, 7.83)	7.0 (3.0, 11.2)		0.93

## Data Availability

The data presented in this study are available on reasonable request from the corresponding author. The data are not publicly available due to potential privacy issues.

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
