# Peer review of "Deep Learning Approach Predicts Longitudinal Retinal Nerve Fiber Layer Thickness Changes†"

_bioengineering, 2025, doi:10.3390/bioengineering12020139_

Round 1

Reviewer 1 Report (Previous Reviewer 1)

Comments and Suggestions for Authors

I have examined your study titled "Deep Learning Approach Predicts Longitudinal Retinal Nerve Fiber Layer Thickness Changes" in detail. I have listed the points I found lacking in the article. I believe that the article will reach a better point with a successful revision. The aim of the article is explained as developing deep learning models in the first two lines of the Abstract section. However, 3 machine learning and 1 deep learning architecture are included in the study. These machine learning methods are frequently used methods in the literature. In this section, the innovative aspects of the article should be highlighted. The advantages and disadvantages of these studies should be included in the related studies section. A paragraph about the organization of the article should be added at the end of the Introduction section. The point I particularly want to emphasize is that the methods used in the study are frequently used methods in the literature. Please emphasize the innovative aspects of the article. The Conclusion section should be detailed. Spelling and grammar errors should be reviewed.

Author Response

We would like to thank the reviewer for their time and consideration of our manuscript. Based on reviewer feedback, we have updated the Abstract, Introduction, Discussion, and Conclusion. Our point-by-point responses are provided below in blue and excerpts from the revised manuscript are in red. We thank the reviewer again for their comments and believe that they have helped strengthen our manuscript.

I have examined your study titled "Deep Learning Approach Predicts Longitudinal Retinal Nerve Fiber Layer Thickness Changes" in detail. I have listed the points I found lacking in the article. I believe that the article will reach a better point with a successful revision. The aim of the article is explained as developing deep learning models in the first two lines of the Abstract section. However, 3 machine learning and 1 deep learning architecture are included in the study. These machine learning methods are frequently used methods in the literature. In this section, the innovative aspects of the article should be highlighted.

We agree and have put additional focus on comparing these methods and highlighting the innovative aspects of our custom deep learning approach.

Abstract (lines 26-30): Our custom models used a novel approach that incorporated longitudinal OCT imaging to achieve consistent performance across different demographics and disease severities, offering potential clinical decision support for glaucoma diagnosis. Patient-level data splitting enhances evaluation robustness, while predicting detailed RNFL thickness provides a comprehensive un-derstanding of structural changes over time.

The advantages and disadvantages of these studies should be included in the related studies section.

We have updated the Discussion section to highlight the advantages of our approach compared to other recently published works.

Discussion (lines 331-349): While several previous studies have used AI and DL models for glaucoma diagnosis and progression detection, the majority have focused on cross-sectional data, i.e., using OCT images from a single time point to classify disease status, predict visual function or RNFL thickness.(2,12,29,30) Thus, these models have generally been limited to diag-nosing current disease rather than forecasting future structural changes. In contrast, our study focuses on predicting longitudinal changes in RNFL thickness, offering a more forward-looking approach to disease management. Only one other study has attempted to forecast future RNFL thickness using longitudinal data. Sedai et al.(32) developed a model to predict global and sectoral cpRNFL thickness based on three prior visits, incorporating 3D-OCT data, previous RNFL values, clinical test results, and demographic information, achieving MAEs between 1.8 ± 1.8 μm and 3.1 ± 2.5 μm for healthy and glaucoma pa-tients, respectively. In contrast to our approach, their approach involved training on the first 'N' visits and testing on later visits from the same patients, raising concerns about potential data leakage, as data from a given patient could appear in both the training and test data. Our study used a strict patient-level data split, ensuring no overlap between training and test sets. Additionally, while Sedai et al. predicted only five regional values, our 1D-CNN model forecasts the entire 768-element RNFL thickness vector, offering a more granular resolution to predict regional changes, and potentially reveal clinically relevant patterns that global or sectoral averages might overlook.

A paragraph about the organization of the article should be added at the end of the Introduction section.

We agree and have added a paragraph detailing the organization of our article to the Introduction.

Introduction (lines 98-101): The rest of the paper is organized as follows. Section 2 describes our datasets, pre-processing, and predictive models. Section 3 presents the results of evaluating our pre-dictive models, Section 4 discusses these results and places them in context with other recent work, and Section 5 concludes the work.

The point I particularly want to emphasize is that the methods used in the study are frequently used methods in the literature. Please emphasize the innovative aspects of the article. The Conclusion section should be detailed. Spelling and grammar errors should be reviewed.

We agree with the reviewer. In addition to the changes described above to highlight the innovative aspects of our work, we have also updated the Conclusion section. We have added detail and, again, highlighted our innovations. We have also proofread the manuscript again to remove spelling and grammar errors throughout.

Conclusion (lines 373-382): In summary, this study demonstrates the utility of deep learning models, particularly our custom 1D-CNN approach, in predicting longitudinal RNFL thickness changes in glaucoma patients. Our models were able to accurately predict RNFL changes across patient demographics (race and age) as well as disease severity. They were also able to provide high-resolution (i.e., 768-points) predictions surrounding the ONH. The ability to forecast future RNFL thinning has significant clinical implications for earlier detection and intervention in glaucoma management. While additional work is needed to optimize model performance in specific retinal sectors and validate its applicability in broader clinical settings, our results represent a promising step toward more proactive and personalized glaucoma care.

Reviewer 2 Report (Previous Reviewer 2)

Comments and Suggestions for Authors

No further comments. 

Author Response

No additional comments to address. We'd like to thank the reviewer for their time and consideration of our manuscript.

Round 2

Reviewer 1 Report (Previous Reviewer 1)

Comments and Suggestions for Authors

Congratulations on the successful revision.

This manuscript is a resubmission of an earlier submission. The following is a list of the peer review reports and author responses from that submission.

Round 1

Reviewer 1 Report

Comments and Suggestions for Authors

I have examined your study titled "Deep Learning Approach Predicts Longitudinal Retinal Nerve Fiber Layer Thickness Changes" in detail. I have listed the points I found lacking in your article in bullet points so that your article can reach a better point. The abstract section mainly includes numerical values. The advantages of detecting RNFL should be included in the abstract section, and the study's contributions to the literature should be highlighted. The study is mentioned at the end of the introduction section. However, this section should be expanded by highlighting the innovations and contributions of the article to the literature. The word "photo" in the study should be updated to "image." I want to state that the dataset section is not at the desired level. When I examined it as a researcher, I had difficulty understanding the data format, numbers, etc. In addition, class numbers, etc., should be added. The models used in the study only have names. How were the data given to these models? How were the features of the images extracted? How did you give the OCT images to 1D-CNN networks? Although I have done many studies on the subject, I would like to state that I have had a hard time understanding an article after a long break. Figures explaining how the models work should be added to the study. To summarize, the materials and methods section of the article should be expanded. Details about the dataset used should be provided, and figures should support the study. The article's innovations should be highlighted.

Reviewer 2 Report

Comments and Suggestions for Authors

Queries/Concerns/Suggestions:

·         Suggest authors to add epidemiological statistics for Glaucoma in introduction section.

·         Line#41-44, Author should use full term (abbreviation) for “optic nerve head (ONH)” at first mention subsequently use abbreviation. Line #79: use full term for VF.

·         Authors should mention about exclusion of approx. 65 eyes images from analysis (251*2=502 vs 437) in method section.

·         Among total 251 participants, how many of them have history of diabetes. It would be great if authors cloud compares their model performance in diabetic vs non-diabetic.

·         Authors should correct spacing errors, specifically while citing references.

·         In the result section, considering readers’ perspectives, authors should avoid mentioning too many numbers.

·         In Figure 4: authors should increase font size for number on X-Y axis.

·         Suggest including detailed figure legends.